# Molecular Mechanisms of Microbiota-Mediated Pathology in Irritable Bowel Syndrome

**DOI:** 10.3390/ijms21228664

**Published:** 2020-11-17

**Authors:** Yoshiyuki Mishima, Shunji Ishihara

**Affiliations:** Department of Internal Medicine II, Shimane University Faculty of Medicine, Izumo 693-8501, Japan; si360405@med.shimane-u.ac.jp

**Keywords:** PI-IBS, SIBO, histamine, serotonin, short-chain fatty acids, bile acids

## Abstract

Irritable bowel syndrome (IBS) is one of the most prevalent functional gastrointestinal disorders, and accumulating evidence gained in both preclinical and clinical studies indicate the involvement of enteric microbiota in its pathogenesis. Gut resident microbiota appear to influence brain activity through the enteric nervous system, while their composition and function are affected by the central nervous system. Based on these results, the term “brain–gut–microbiome axis” has been proposed and enteric microbiota have become a potential therapeutic target in IBS cases. However, details regarding the microbe-related pathophysiology of IBS remain elusive. This review summarizes the existing knowledge of molecular mechanisms in the pathogenesis of IBS as well as recent progress related to microbiome-derived neurotransmitters, compounds, metabolites, neuroendocrine factors, and enzymes.

## 1. Introduction

Irritable bowel syndrome (IBS) is a chronic gastrointestinal disorder characterized by recurrent symptoms including abdominal pain and abnormal bowel habits in the absence of an apparent organic cause [1,2]. It is one of the most prevalent diseases seen in patients visiting gastrointestinal (GI) clinics and global prevalence is estimated to be approximately 10–15%, with a range of 1.1–35.5% reported worldwide [3]. Questionnaires regarding abdominal pain and bowel habits based on the Rome IV criteria are widely used for diagnosis of IBS, as well as to further characterize subtypes based on fecal condition and bowel movements, termed diarrhea-predominant (IBS-D), constipation-predominant (IBS-C), mixed (IBS-M), and unclassified (IBS-U) [4,5]. Current treatments for IBS selected in medical practice are generally symptomatic therapies with limited efficacy [6,7,8]. Clinicians have been reported to occasionally encounter a refractory case in which irregular symptoms markedly decrease quality of life, social activities, and productivity of the affected patient [2,9]. Because of the chronic features, patients with IBS place a huge economic burden on healthcare systems [10,11].

Several studies have been conducted to seek better treatment strategies for IBS. However, that goal remains challenging because it can be difficult to obtain quantified reproducible data for investigating functional disorders. IBS is easily influenced by psychological factors, with significant placebo effects often appearing in clinical studies, making the results difficult to interpret [1,12]. Questionnaires but not specific examinations are used to diagnose IBS based on standard criteria [3,4,5], indicating that prevalence analysis may not always reflect the actual dynamics of IBS patients. Furthermore, there are no useful biomarkers or reliable tests available to evaluate disease activity [5,13]. An additional issue is the heterogeneous characteristics of IBS, with multiple factors, including age, gender, ethnicity, culture, socioeconomic status, and geographical location, having an influence on its prevalence [1,2,3,14]. As a result, findings obtained in clinical studies can vary due to those factors along with the significant placebo effects [3]. Finally, no appropriate animal models that fully reflect IBS in humans are available. Although several models have been proposed, there are large differences between human and animal models of IBS related to the resident enteric microbial community, mucosal immune system, diet, sociality, and psychological status [15,16]. Breakthrough research results that deal with these multiple issues and the huge clinical demand are highly anticipated.

When considering methodologies to identify unknown mechanisms related to IBS, microbiological approaches appear to have great potential [17,18]. It is well known that enteric microbiota reside in the intestines and communicate for mutual benefit [19,20]. Although classical culture methods are unable to detect the full varieties of resident organisms in human intestinal tissues, recent advances in microbiological technologies, including metagenomics, metabolomics, and metatranscriptomics, have provided the identification of whole microbial profiles, as well as findings that show microbial functions and host interactions, with abundant novel observations reported [21,22,23].

In the present review, existing knowledge and recent progress in understanding of the molecular mechanisms involved in the gut microbiota-mediated etiology of IBS are collectively summarized. Our intention is to shed light on the importance of further IBS research targeting enteric microbiota, which will provide a deeper understanding regarding the pathogenesis of this disease.

## 2. Role of Intestinal Resident Microbiota in IBS Pathology

The human GI tract harbors trillions of microorganisms that influence both health and disease development [19,24]. This community of intestinal microbiota is dependent on host genetics and environmental factors, and is fundamentally stable over time in most individuals [20,25]. Resident microbiota defined as “healthy” have been shown to provide beneficial effects for the host with multiple mechanisms [26,27]. These microbes produce key enzymes and metabolites that help to absorb essential nutrients and vitamins [28], and are also important for development and function of the mucosal immune system, which is required to provide a prompt and effective response to pathogens while remaining tolerant of harmless food antigens or other commensal species [26,29]. Another essential role of this microbiota is pathobiont colonization resistance by filling distinct niches, exploiting oxygen, changing pH levels, outcompeting for nutrient resources, secreting transmitters, and inducing host immune activation [30,31]. Together, healthy resident microorganisms prevent invasive pathogens and maintain gut microbial ecosystems, which are important for the wellbeing of the host.

On the other hand, pathogenic species exist in normal resident microbiota as well [32,33]. These are presumed to be minor species and oppressed by beneficial ones in a healthy gut condition, though they increase and become activated when the intestinal environment undergoes changes, such as by usage of antibiotics, an altered diet and/or lifestyle, and exposure to severe enteritis [32,34]. Such imbalance in the microbial community is termed “dysbiosis” and potentially has harmful effects on mucosal homeostasis [35,36]. Indeed, dysbiosis is often observed in a variety of intestinal disorders including IBS [17,35,37]. For example, the small intestinal bacterial overgrowth (SIBO), characterized by the overgrowth of colonic type bacteria, such as *Klebsiella, Enterococcus*, and *Escherichia*, is frequently observed in the small intestine of IBS patients [38,39]. SIBO can potentially induce increased gut permeability, low grade inflammation, dysmotility, and decreased absorption of bile salts, and also alter enteric nervous system (ENS) activity [38]. In addition to those bacteria noted above, overgrowth of enteric archaea species also causes IBS symptoms. For example, *Methanobrevibacter smithii* produces abundant methane gas in the small intestine, influencing gut motor activity and delaying bowel transit, and is potentially associated with IBS-C [40,41].

Over the past two decades, microbiome research has rapidly progressed with novel technologies introduced, including whole genome shotgun sequencing and multi-omics analysis [23,42,43], which help to reveal IBS-specific dysbiosis [37]. A relatively common observation is that microbial diversity in fecal samples from IBS patients is lower than in those from healthy control subjects [17,36,44]. Additionally, the ratio of *Firmicutes*/*Bacteroidetes*, a rough indicator of a microbial composition shift, is higher in IBS patients in most but not all related studies [45,46], suggesting that more in-depth analysis is required for greater specificity. A meta-analysis found lower levels of fecal *Lactobacillus* and *Bifidobacterium*, along with higher levels of *Escherichia coli* and *Enterobacter* in feces samples obtained from IBS patients [47]. In a systematic review, the *Proteobacteria* phylum, *Enterobacteriaceae* and *Lactobacillaceae* families, and *Bacteroides* (phylum *Bacteroidetes*) genus were increased in such patients as compared with controls [46], whereas uncultured *Clostridiales* I, and the *Faecalibacterium*, and *Bifidobacterium* genera were decreased in association with IBS [48]. Also, the downregulation of bacterial colonization, including that by *Lactobacillus, Bifidobacterium*, and *Faecalibacterium prausnitzii*, has been observed in IBS cases, particularly IBS-D [49]. However, it remains to be determined whether a specific dysbiosis occurs in all affected patients, or even whether such a condition is a cause or consequence of IBS. The former is supported by evidence presented in several reports showing the features of human IBS, such as visceral colonic hypersensitivity can be transferred to a germ-free animal by fecal transplantation [50], development of post-infectious IBS (PI-IBS) after severe colitis with dysbiosis [51,52], and improved IBS symptoms by reversing dysbiosis [8,53], as well as microbial physiology alterations that can cause IBS symptoms, which are described in the present review. Findings related to the latter are supported by results showing that abnormalities in bowel transit and stool consistency in IBS patients alter gut microbial richness, composition, and metabolic profile [54,55], and that low-grade mucosal inflammation in affected patients can also alter enteric microbial composition [56].

Treatment strategies targeting enteric microbiota, including antibiotics, pre-/pro-/symbiotics, a low-FODMAP (fermentable oligo-, di-, mono-saccharides, and polyols) diet, and fecal microbiota transplantation (FMT), have been found to be effective in certain populations [45,57]. Particularly, FMT appears to be one of the most powerful modalities to reverse dysbiosis and potentially induces long-lasting remission of IBS [58,59]. Recent study demonstrated that utilizing a well-defined donor with a normal dysbiosis index and favorable specific microbial signature is important for successful FMT in IBS patients [60]. Whereas the efficacy of FMT is not always superior to placebo [61] and several safety concerns including fatal infection have been raised [62,63]. More recently, potential risk of transmission of severe acute respiratory syndrome coronavirus 2 (SARS-CoV-2) via FMT has been considered [64]. Therefore, careful consideration of FMT should be given in case of IBS. Those multiple preclinical and clinical observations highlight the importance of dysbiosis for a better understanding of IBS pathology. However, findings among studies presented thus far appear to be inconsistent, which may be due to several of the issues described earlier and/or differences among analytic techniques. Fortunately, new technical approaches including metabolomic analysis have allowed for a more precise identification of the microbiota-mediated pathogenesis of IBS [65,66].

Studies that utilized germ-free animals have indicated that resident gut microbiota influence intestinal motility and pain sensation. Such animals display a significantly delayed gut transit and a prolonged phase of interdigestive migrating motor complex [67], while microbiota-mediated gut motility appears to depend on the type of microorganisms present. Inoculation with either *Lactobacillus acidophilus* or *Bifidobacterium bifidum* was shown to increase activity of that complex as well as bowel transit time, whereas colonization with *E. coli* and *Micrococcus luteus* resulted in delayed gut motility [68]. Of interest, germ-free rats inoculated with fecal materials from IBS patients developed stronger visceral hypersensitivity as compared with rats colonized with such materials from healthy controls [50], strongly suggesting that dysbiosis is a cause of GI dysfunction. Nevertheless, interpretation of results from studies that used germ-free animals must be performed with caution because such animals have an immature mucosal immune system, barrier system dysfunction, insufficient neurodevelopment, and altered psychological condition [37,69,70].

PI-IBS is a type of IBS that develops after severe enterocolitis, with the term developed based on findings showing that related symptoms arise de novo following exposure to acute gastroenteritis and then remain sustained despite clearance of the provoking pathogen [51,71,72]. Bacteria including *Escherichia coli*, *Campylobacter jejuni*, and *Salmonella* spp., as well as viruses (*norovirus*, *cytomegalovirus*) and parasites (*Blastocystis*, *Cryptosporidium*, *Giardia*) have often been identified as causative microorganisms in PI-IBS cases [51,72,73]. The proposed pathogenesis is persistent low-grade mucosal inflammation with activated immune cells and increased mucosal permeability following dysfunction of the epithelium barrier [51,72], which are presumably associated with microbial dysbiosis after enterocolitis. Of note, clinical features and fecal microbial profiles shown in PI-IBS are similar to those in IBS-D cases [74,75], indicating that investigation of PI-IBS can provide insight into the pathophysiology of IBS-D. An advantage of studying PI-IBS is reproducible data, because of the clearly defined onset and pathobiont, making it easier to identify causal relationships [16]. As a result, multiple studies have attempted to unveil related molecular mechanisms by analyzing animal models of PI-IBS and human patients, with novel insights into the microbiome-mediated pathophysiology of IBS presented [14,16,35,72].

## 3. Mechanisms of Action: Influence of Microorganisms on Intestinal Homeostasis and IBS 

Accumulating evidence has revealed bidirectional communication of intestinal microbiota with the ENS and central nervous system (CNS), termed the “brain-gut-microbiome axis” [37,70,76,77]. This concept suggests that gut microbiota can control human brain activity through the ENS, indicating that resolution of dysbiosis is a reasonable treatment strategy for IBS. This review focuses on gut microbe-derived neurotransmitters, compounds, metabolites, enzymes, and endocrine factors. Each of those is potentially altered by dysbiosis and involved in the pathology of IBS, such as visceral hypersensitivity, increased intestinal motility, mucosal immune activation, and epithelial barrier dysfunction. (Figure 1 and Figure 2, Table 1)

### 3.1. Microbial Neurotransmitters

Microbiota-derived neurotransmitters play an important role in IBS symptom development, especially visceral pain [124,142]. As noted in those studies, most are synthesized in the intestine, and have effects on the ENS and then the CNS, or vice versa [124,142]. Key neurotransmitters that serve a fundamental role in the pathology of IBS are described following.

#### 3.1.1. Histamine

Histamine is a biological amine that is important for essential physiological functions, including cell proliferation/differentiation, hematopoiesis, and regeneration [78,80]. It also affects a variety of immune reactions related to allergy and inflammation, modulation of GI motility, an increase in mucosal permeability, and alteration of mucosal ion secretion, which are potentially associated with the pathogenesis of IBS [53,81,82]. Supernatants from IBS colonic samples have been found to contain increased levels of histamine and shown to stimulate submucosal neuronal activity in rats [83]. In addition, expression levels of the histamine receptors H1R and H2R in intestinal tissue samples from IBS patients are upregulated [84], while administration of antagonists for histamine receptor H1R have been reported to improve abdominal pain in IBS patients as well as hypersensitivity in animal models of IBS through the transient receptor potential vanilloid 1 (TRPV1) [79]. Histamine is synthesized by host cells, including mast cells and basophils, while *E. coli* and *Morganell morganii*, resident gut microbial species, have been identified as histamine-producing bacterial strains [78,187]. Certain bacteria also enable production of histidine decarboxylase (HDC), which converts the precursor molecule histidine to histamine [188]. It is therefore considered that dysbiosis with increased histamine- or HDC-secreting bacteria is potentially associated with development and aggravation of IBS.

#### 3.1.2. Serotonin

Serotonin (5-hydroxytryptamine, 5-HT) is a multifunctional transmitter biosynthesized mainly in small intestinal enterochromaffin cells in association with the metabolism of tryptophan [86,189]. Resident bacteria, including *Corynebacterium* spp., *Streptococcus* spp., and *Enterococcus* spp., also directly produce 5-HT [190], while indigenous spore-forming bacteria regulate host serotonin biosynthesis [191]. Enteric 5-HT appears to be one of the key molecules in the pathology of IBS through increasing epithelial permeability, visceral hypersensitivity, activation of immune cells, and changes in motility, all of which corporately induce IBS symptoms [85,86]. Currently, antagonists of the serotonin receptors 5-HT3 and 5-HT4 are widely utilized in clinical practice for treating IBS [7,88]. Certain patients with IBS have a significant decrease in expression of the serotonin transporter (SERT), which causes an increased level of mucosal 5-HT and triggers IBS symptoms [192,193]. Susceptible genes in serotonergic signaling pathways have been discovered in genome wide association studies (GWAS) [85,194]. Of considerable interest, enteric 5-HT was found to play an important role in quorum sensing (QS), a density-dependent signaling mechanism in microbiota, while 5-HT stimulates a QS-dependent virulence factor and biofilm production in pathobionts, and expands infection [87]. In a normal condition, members of enteric microbiota have mutual interactions to shape a stable microbial complex by 5-HT-mediated QS, whereas a dysfunction of the 5-HT pathway can potentially cause dysbiosis in the IBS. Accordingly, dysbiosis conditions can be either a cause or consequence of the aberrant activation of serotonin signaling in IBS development.

#### 3.1.3. Glutamate

Glutamate, an excitatory neurotransmitter in both host CNS and ENS, is mainly derived from diet [92]. The peripheral N-methyl-d-aspartate (NMDA) receptor, an ionotropic receptor of glutamate, is important for visceral pain transmission [89], while dysfunction of glutamatergic signaling is associated with a variety of neuropathological disorders [93,94,95] as well as IBS through the brain-gut-microbiome axis [90,92]. Indeed, dietary glutamate worsens symptoms of IBS along with fibromyalgia [96], and an activation of colonic NMDA receptor induces the visceral hypersensitivity in IBS by increasing production of brain-derived neurotrophic factor [97]. Glutamate appears to be involved in the mechanism of intestinal and cerebral inflammation responses despite decreasing intestinal permeability [98,99]. Further study will be needed for these observations. Gut microbiota, including *Bacteroides vulgatus* and *Campylobacter jejuni*, have effects on glutamate metabolism [93] and certain lactic acid bacterial strains are able to produce glutamate [195]. The soil bacterium *Corynebacterium glutamicum* is known as one of the most powerful of glutamate-producing bacteria and widely utilized in industrial applications [196]. In addition, enteric bacteria use glutamate in nitrogen metabolism as well as protection from a variety of factors, such as endoplasmic reticulum (ER) stress [91]. Thus, glutamate influences multiple biological processes and the survival of gut microbiota and is potentially involved in shaping dysbiosis in the gut.

#### 3.1.4. γ-Aminobutyric Acid (GABA)

In contrast to glutamate, GABA is an inhibitory neurotransmitter that regulates perception of visceral pain from peripheral afferent neurons [100]. GABA is synthesized from glutamate via glutamate decarboxylase enzymes in the diet, while several bacterial strains such as *Lactobacillus brevis* and *Bifidobacterium dentium* can also synthesize that in the intestines [101,197]. GABAergic signaling is involved in the pathogenesis of multiple CNS disorders and known as a therapeutic target [102]. In the context of IBS, Aggarwal et al. demonstrated that reduced GABA concentration and an altered GABAergic signal system contribute to the pathogenesis of IBS-D, potentially by alleviating inflammatory processes [103]. Also, the administration of GABA-producing *Bifidobacterium dentium* as a probiotic was reported to modulate intestinal visceral hypersensitivity in an animal model of IBS [104]. Together, an imbalance between gut dysbiosis-mediated glutamatergic vs. GABAergic signaling appears to play a critical role in ENS/CNS activity, and is likely involved in the pathogenesis of IBS [92,101,103,105].

#### 3.1.5. Others

The essential neurotransmitters Noradrenaline (NA), dopamine (DA), and acetylcholine (ACh) are also produced by enteric bacteria [112,119,198], and have influences on both the psychological and physiological features of IBS.

NA, a chemical class of catecholamine, is an essential neurotransmitter in multiple physiological processes, such as the cardiovascular and endocrine systems, as well as CNS activity [106,107]. Intrathecal administration of NA into the spinal cord was found to induce propulsive contractions of the colorectum through α1 adrenoceptors [108]. NA shows a pleiotropic function in mucosal inflammation. In low concentrations, NA plays a pro-inflammatory role through the α-adrenergic receptors, whereas in higher concentrations, it has more affinity for the β-adrenergic receptors with its anti-inflammatory effects [111]. Serotonin-noradrenaline reuptake inhibitors have also been shown effective for IBS patients, especially those with IBS-D, and also animal models [109,110].

DA is a precursor for NA and adrenaline, and functions in a reward-motivated manner in the CNS [112]. DA is associated with perception of chronic pain, pain-induced anxiety and depression [116], while it controls systemic inflammation through inhibiting NLRP3 inflammasome and may be involved in intestinal inflammation [117,118]. DA also has been shown to adjust contraction and relaxation in the small intestine [113]. The dopamine D2 receptor antagonist stimulates GI motility and has been utilized as an antiemetic based on its ability to relieve GI symptoms in certain IBS patients, despite a lack of firm clinical evidence [115]. In addition, DA may enhance duodenal epithelial permeability via the dopamine D5 receptor, while it serves as mucosal protection by promoting colonic mucus secretion through the same receptor [114]. These findings indicate that dopaminergic signaling can influence IBS symptoms. Moreover, several pathogenic bacteria, including *Escherichia coli O157:H7 (EHEC)*, *Klebsiella pneumoniae, Pseudomonas aeruginosa, Shigella sonnei*, and *Staphylococcus aureus*, utilize DA or norepinephrine for growth and function [112], suggesting that susceptibility to those pathogens may be determined by transmitters and that alternation of the dopaminergic pathway can induce dysbiosis.

The bioactivity of ACh is regulated by two major enzymes, acetyltransferase (ACh-synthesizing enzyme) and acetylcholinesterase (ACh-degrading enzyme) [112,119]. The mucosal immune system uses these enzymes to regulate the balance between ACh and catecholamines in response to gut microbiota [112,119]. In addition, some gut microbiota are capable of secreting ACh and likely affect gut immunity via this system [111]. A previous study showed that serum cholinesterase activity was increased in female IBS-D patients [120], suggesting involvement of a dysfunction of the AChergic signal pathway caused by enteric dysbiosis in the pathogenesis of IBS.

Collectively, NA, DA, and ACh appear to function as communication tools between microbes, the mucosal immune system, and brain activity, and thus dysbiosis may result in their miscommunication [112].

### 3.2. Microbial Compounds

Enteric microbiota are critical components for developing and maintaining the intestinal immune system, which prevents establishment of pathogenic organisms and maintains health [26,29,199]. A key process for eliminating pathogens begins by sensing specific patterns of microbial components by Toll-like receptors (TLRs) [121,122]. Activated TLR signaling stimulates not only cytokine production through the innate immune system, but also directly affects visceral pain and gut motility as part of the IBS pathology [122,200].

#### 3.2.1. Toll-Like Receptor Ligands

TLRs are evolutionarily preserved pattern-recognition receptors that play an important role in activation and regulation of the gut innate immune system [121,122]. IBS patients possess elevated expression of TLRs with innate immune cell activation in intestinal mucosa, with TLR2 known to recognize lipoproteins derived from various bacteria, while TLR4 and TLR5 recognizes bacterial lipopolysaccharide (LPS) and flagellin, respectively [201,202,203]. LPS released from pathogenic microorganisms can increase gut permeability, and alter activities of the ENS and CNS [123]. In IBS-D patients, serum LPS levels and anti-flagellin antibodies are elevated, and known to be associated with anxiety score [204]. Expression of TLR9, which recognizes the unmethylated CpG motif, is also increased in IBS and single nucleotide polymorphisms (SNPs) for TLR9 were discovered in a Canadian cohort study targeting PI-IBS [205]. Those observations indicate aberrant activation of TLRs in a certain group of affected patients, which is associated with dysbiosis and IBS development.

#### 3.2.2. Cytokines

After sensing particular microbial patterns by the TLRs, several inflammatory and regulatory cytokines are produced by host cells, which can influence mucosal permeability and visceral pain in IBS patients [52,124]. Colonic soluble mediators noted in animal models of IBS were shown to activate submucosal neurons via an interleukin (IL)-6-dependent mechanism [125]. IL-6, IL-1β, and tumor necrosis factor α (TNF-α) have been found to have direct effects on neuronal activity in mice, which in turn altered gut contractility, absorption, and/or secretion [126]. Additionally, IL-6 was reported to increase epithelial permeability by influencing mucosal ion transport [127]. Moreover, those proinflammatory cytokines are potent modulators of corticotropin-releasing hormone (CRH), which stimulates activity of the hypothalamic–pituitary–adrenal axis (HPA), as well as increases in ACTH and cortisol [206]. Alternation of this pathway is associated with psychological disorders such as depression and also visceral hypersensitivity in IBS [206,207,208]. In plasma samples from IBS patients, elevated levels of pro-inflammatory cytokines, such as IL-6, IL-8, IL-1β, and TNF-α, and with decreased regulatory IL-10, have been noted [128,129]. Also, soluble mediators found in colonic biopsy samples from IBS patients were shown to stimulate enteric neuronal activation, while plasma samples from such patients stimulated rat submucosal neurons in an IL-6-dependent manner [130]. Furthermore, cytokine modulates GI motility in multiple gastrointestinal disorders including IBS [209]. Finally, IBS-related gene polymorphisms have been identified in IL-6, IL-8, IL-10, and TNF, as well as TNF superfamily (TNFSF) members [194]. Those observations indicate an aberrant mucosal immune response associated with dysbiosis involved in the pathogenesis of IBS.

#### 3.2.3. Other Microbial Products

Some pathogenic bacteria are able to produce pore-forming toxins [131]. Such toxins increase mucosal permeability and compromise mucosal barriers, and also play a role in visceral pain by depolarizing nociceptive neurons and compromising epithelial integrity [124,133]. N-formylated peptides are produced by certain pathogenic bacteria [131], and involved in visceral pain sensation by releasing inflammatory mediators and provoking neural activation through formyl peptide receptors [124]. *Staphylococcus aureus* directly activates nociceptor neurons by N-formylated peptides and the pore-forming toxin α-hemolysin [132]. Although scientific evidence is still limited, dysbiosis increased by these microbial products appears to influence low-grade inflammation-mediated nociception in IBS.

### 3.3. Microbial Metabolites

Microbiota members produce metabolites as part of biological and enzymatic processes, which influence a variety of physiological factors in host intestines [210,211]. An altered enteric metabolic profile of either the host or microbiota, or their interactions are likely associated with IBS symptoms [212]. Key microbial metabolites in the pathogenesis of IBS including tryptophan, short-chain fatty acids (SCFAs), bile acids, and vitamins are described in the following sections.

#### 3.3.1. Tryptophan

Tryptophan is an essential amino acid that has a fundamental role in the brain-gut-microbiome axis through its three major metabolic pathways; the 5-HT (previously described), kynurenine, and indole-aryl hydrocarbon receptor (AhR) pathways, the latter of which is tightly modulated by intestinal microbiota [124,134].

In the human body, only 1% of ingested tryptophan is converted into 5-HT, while the majority is catabolized by indoleamine 2,3-dioxygenase (IDO1) into kynurenic and quinolinic acid [213]. The kynurenic pathway shows a pleiotropic effect in the pathophysiology of IBS. Kynurenic acid has a regulatory function in both the CNS and GI tract via the G protein-coupled receptor GPR35, which potentially ameliorates inflammatory-mediated nociception in IBS [134,135]. Decreased kynurenic acid as well as 5-HT levels have been observed in the intestinal mucosa of IBS patients [214]. On the other hand, both kynurenic and quinolinic acid can bind the NMDA receptor located on neuronal networks in the spinal cord, which is involved in peripheral sensitization and visceral hyperalgesia [89]. Indeed, quinolinic and kynurenic acid production is activated with increased expression of the NMDA receptor 2B in IBS-D patients [215]. Additional examinations regarding this pathway in IBS are needed.

Microbiota members such as *E. coli*, *Achromobacter liquefaciens*, and *Bacteroides* spp. metabolize dietary tryptophan into indole, which has beneficial effects on host intestinal functions through the AhR [216]. AhR-mediated signaling modulates mucosal homeostasis by several mechanisms, such as enhancement of mucosal defense, activating Th17 cells and neutrophils, modulation of mucosal immunity by IL-22 production and regulatory T cell induction, strengthening the epithelial barrier function by promoting goblet cell differentiation and mucus production, and maintaining intestinal stem cell homeostasis [136,137,138,139]. Probiotic bacteria such as *Lactobacillus reuteri* and *Lactobacillus bulgaricus* can activate this pathway to regulate inflammation and correct dysbiosis [138]. Moreover, similar to 5-HT, bacteria-produced indole functions as a QS signal that regulates virulence of and biofilm formation by bacteria, indicating it as an important factor for maintaining microbial composition [140,141].

Tryptophan together with its metabolites influences a variety of bioactivity factors as well as microbes in the host, which can be impaired by gut dysbiosis.

#### 3.3.2. Short-Chain Fatty Acids

SCFAs, mainly synthesized by microbial fermentation of dietary starches in the colon, have an essential role in intestinal homeostasis, such as harvesting energy from colonocytes, barrier function, wound healing, immunocyte regulation, host metabolism, and nociception [143,147], and an SCFA metabolism abnormality is likely involved in the pathogenesis of IBS [37,70,148,149]. IBS-C patients have been shown to have increased levels of colonic acetate, propionate, and total organic acids along with increased *Veillonella* and *Lactobacillus* [217], while SCFAs can accelerate colonic transit by stimulating 5-HT release and associated secretory effects, potentially contributing to IBS-D [150,151]. Butyrate, one of the major SCFAs, plays an anti-inflammatory role by inducing regulatory immune cells, and modulating colonocyte proliferation and apoptosis [152], with controversial results showing effects on visceral functions involved in the pathology of IBS reported [142]. Butyrate was found to decrease visceral pain in healthy human volunteers as well as colonic hyperpermeability in rat models of IBS [153,154], while rectal administration aggravated visceral pain in normal rats and those with colitis [144,145]. The latter result may have been due to a nerve growth factor that induces butyrate-mediated colonic hypersensitivity in an inflammation-independent manner [146]. SCFAs can also influence neuroendocrine function. Acetate, butyrate, and propionate stimulate free fatty acid receptor 2 (FFAR2)/G-protein–coupled receptor 43 (GPR43) and FFAR3/GPR41 in L cells in the distal ileum to secrete neuropeptides including glucagon-like peptide-1 (GLP-1) and peptide YY (PYY) [218,219,220], indicating that SCFA-FFA signaling is likely important for gut physiological functions.

#### 3.3.3. Bile Acids

Primary bile acids (cholic acid, chenodeoxycholic acid) are synthesized from cholesterol in the liver and excreted into the intestinal lumen where gut microbiota convert excreted primary bile acids into secondary bile acids [70,221]. The pool of bile acids is tightly controlled by the farnesoid X receptor (FXR)-dependent pathway in the liver and ileum, and conjugated bile acids released into the intestine are reabsorbed in the distal ileum and returned to the liver through the portal vein [222]. Only a small amount of bile acids flow into the colon via the bile acid recycling system under a normal condition. On the other hand, excess delivery of bile acids into the colonic lumen by a variety of pathogenic conditions, including dysbiosis, FXR pathway dysregulation, and bile acid transporter dysfunction, causes net fluid and electrolyte secretion, resulting in a predominance of diarrhea [155,156]. Hence, the regulation of dysbiosis-mediated abnormal bile acids in the GI tract is likely involved in the pathogenesis of IBS-D [157]. Bile acid malabsorption was shown to occur in about 30% of IBS-D patients, while administrations of adsorption drugs for bile acids including cholestyramine were effective in those patients [158,159]. Zhao et al. demonstrated that *Clostridia*-rich gut dysbiosis is positively correlated with fecal bile acid concentration as well as increased bile acid synthesis/excretion in IBS-D patients [160]. A recent report demonstrate that gut microbiota dysbiosis is associated with bile acid metabolism [223]. Therefore, bile acid sequestrant therapy appears to be a reasonable approach for management of IBS-D, though the sequestrant can interfere with absorption of other IBS drugs, potentially weakening their efficacy [159].

#### 3.3.4. Vitamins

Among vitamins, vitamin D and B6 have been extensively investigated in IBS research studies. Vitamin D plays an important role in maintaining gut homeostasis by enhancing intestinal barrier functions, modulating immune responses, and serving as an antimicrobial agent [161,162], while its deficiency induces proinflammatory cytokines, including TNF-α and IFN-γ, which weaken mucosal barrier function by downregulating tight junction proteins [165,166]. Mice lacking the vitamin D receptor (VDR) exhibited an inflammatory phenotype with increased mucosal permeability, and were found to be susceptible to mucosal damage and colitis [164]. On the other hand, supplementation with 1,25(OH)2D3, an active form of Vitamin D, protects against LPS- or TNF-α-induced barrier dysfunction [167,168]. VDR is essential for butyrate-mediated inhibition of inducible NFκB activation and regulates expression of tight junction proteins [168,169]. In addition, 1,25(OH)2D3 can directly interact with gut microbiota and potentially ameliorate existing dysbiosis. Supplementation with vitamin D3 was found to increase the concentration of 25(OH)D in serum, along with increases in beneficial bacteria and decreases in pathogenic species [224,225]. In contrast, certain pathogenic microorganisms such as *Salmonella typhimurium* can decrease VDR expression in the colon [226], though that can be reversed by probiotics such as *Lactobacillus rhamnosus* strain GG and *Lactobacillus plantarum* [162,227]. Another important property of vitamin D is its antimicrobial function. 1,25(OH)2D3 and VDR interact with Paneth cells and promote secreting antimicrobial peptides, including cathelicidins, β-defensin-2, and lysozyme [163]. Therefore, VDR dysfunction and/or vitamin D deficiency causes dysbiosis and weakens host defense through dysbiosis-mediated low-grade mucosal inflammation and barrier dysfunction [162]. Clinically, VDR is overexpressed in the duodenum of patients with IBS [228]. A low level of vitamin D has been observed in the serum of IBS patients, while vitamin D supplementation was demonstrated to improve disease activity [229]. These findings emphasize the importance of awareness of aberrant vitamin D signaling pathways in refractory IBS patients.

Vitamin B6 is a water-soluble vitamin that is present in several forms [230]. The bioactive form, pyridoxal 5′-phosphate, serves as a coenzyme in a variety of essential cellular processes, including those related to amino acids, carbohydrates, and lipid metabolism [231], while its metabolic pathways are mediated by multiple bacterial species and their enzymes [230,231]. A deficiency of vitamin B6 is likely associated with an inflammatory condition [232], thus it might be involved in the inflammation-mediated pathology of IBS. Indeed, a low intake of vitamin B6 has been shown to be correlated with worsened IBS symptoms, while supplementation with vitamin B6 alleviated severity [170].

Collectively, dysbiosis-mediated metabolic changes in vitamin B6 and D should be considered for effective management of IBS patients.

### 3.4. Microbial Endocrine Factors

Enteroendocrine cells regulate several important GI tract functions, including motility, secretion, absorption, visceral sensitivity, local immune defense, and cell proliferation, as well as appetite [233,234]. As described above, these cells sense microbial metabolites and secrete a variety of hormones such as somatostatin, motilin, cholecystokinin, neurotensin, vasoactive intestinal peptide (VIP), peptide YY, and glucagon-like peptides [233,234]. Following, we focus on glucagon-like peptide-1 (GLP-1) and peptide YY (PYY), which have been intensively investigated in enteroendocrine-related IBS research studies.

#### 3.4.1. Glucagon-Like Peptide-1 (GLP-1)

Intestinal L cells secrete GLP-1 following stimulation with a variety of microbiota-mediated molecules, such as SCFAs, bile acids, and bacterial products, as described earlier [70,220]. GLP-1 plays a role as a signal transducer in the brain-gut-microbiome axis, while it also interacts with the HPA axis and immune system as part of the pathology of IBS or inflammatory bowel disease (IBD) [171,172,174]. Furthermore, this hormone has been shown to increase colonic transit and alleviate visceral pain sensitivity related to IBS-C [171], whereas decreased levels of circulating GLP-1 and mucosal GLP-1 receptor expression have been correlated with disease severity in IBS-C patients [175]. Moreover, GI transit can be modulated by expression of the microbiota-dependent GLP-1/GLP-1 receptor, as gut microbiota accelerate GI motility while suppressing the expression of GLP-1 receptors in myenteric neural cells throughout the GI tract [173]. Therefore, dysbiosis in the gut potentially influences IBS via GLP-1 regulation.

#### 3.4.2. Peptide YY (PYY)

Like GLP-1, PYY is secreted mainly from intestinal L cells by microbial stimulation [172,177]. PYY regulates GI motility and also secretion, absorption, appetite, and visceral sensitivity by altering the release of serotonin from colonic enterochromaffin cells [172,177]. Multiple studies have demonstrated PYY abnormalities in the pathology of IBS, with the concentration of PYY and number of PYY cells found to be decreased in the colon of patients with IBS [176,178]. Additionally, it has been speculated that a low PYY level can contribute to intestinal dysmotility and visceral hypersensitivity in those patients [176]. However, PYY release can also be stimulated by multiple IBS-related mechanisms including neural flex and gut neuroendocrine peptides [172,177], and thus it remains unclear whether a decrease in the PYY pathway is a cause or consequence of dysbiosis in IBS.

### 3.5. Microbial Enzymes: Proteases

An intestinal epithelial barrier protects from invasive pathogens and also preserves the ability to absorb essential nutrients [235,236,237]. The epithelial and mucus layers provide physical separation from luminal microbiota, while mucins, sIgA, antimicrobial peptides, and various types of immune cells corporately reinforce host mucosal defense [235,238,239,240,241,242]. Mucosal barrier dysfunction was found to cause increased permeability, which potentially induces bacterial translocation and triggers a variety of disorders including IBS [243]. Indeed, increased gut membrane permeability is correlated with visceral and thermal hypersensitivity seen in IBS patients [179]. Also, a reduction of tight junction proteins has been observed in association with IBS [180], while CDH1, related to the tight junction protein E-cadherin, was reported to be a susceptible gene for PI-IBS [205].

Microbial factors associated with mucosal barrier function include microbial proteases, as well as neurotransmitters and metabolites, as previously described. Luminal proteases derived from host digestive enzymes and the resident microbiota bind to protease activated receptors (PARs), which are critical for multiple physiological processes [182,244]. PAR-1 and 2 are expressed by enterocytes, as well as immune cells in the GI tract, and influence mucosal permeability, inflammation and motility as part of the pathogenesis of IBS [181,182,183,186]. Additionally, microbe-derived proteases such as gelatinase, gingipains, serralysin, and LasB mainly activate PAR-2, which regulates mucosal permeability through myosin light chain kinase (MLCK) [182]. However, in the pathogenesis of IBS, alternation of these microbial proteases by dysbiosis can allow for a direct attack on epithelial barriers or indirectly induce barrier dysfunction via PARs in host cells [124]. Fecal supernatants from IBS-D patients were shown to enhance paracellular permeability of murine colonic epithelium, while a cocktail of serine protease inhibitors or utilization of colonocytes from PAR-2 knockout mice negated that effect [184]. A more recent study also showed that serine proteases in fecal supernatants from IBS patients disrupted the mucosal barrier through PAR-2-mediated signaling, MLCK phosphorylation, and loss of tight junction proteins [185].

## 4. Conclusions

Accumulating evidence indicates that intestinal dysbiosis is deeply involved in the pathogenesis of IBS and thus an ideal target for treating affected patients, though IBS-specific dysbiosis has not been fully revealed, likely due to the complex pathology of heterogeneous IBS-related conditions. More precise identification of the mechanisms involved will guide further effective and safe microbe-based induction, and lead to maintenance therapies for IBS.

## Figures and Tables

**Figure 1 ijms-21-08664-f001:**
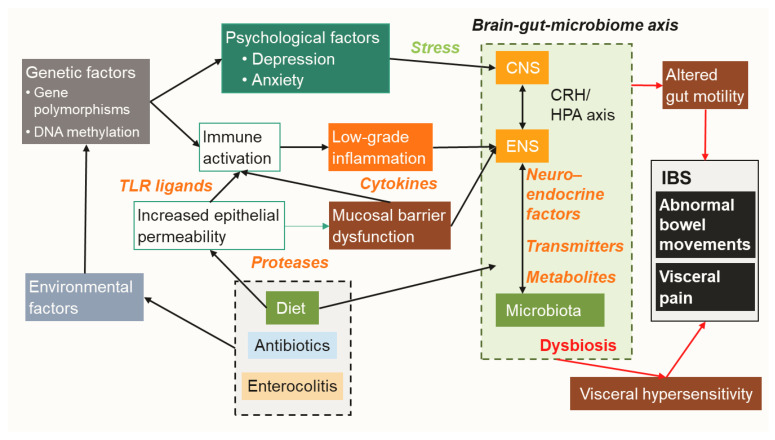
**Overview of pathogenesis related to IBS.** IBS is a multifactorial disorder affected by various genetic and environmental factors. Altered diet, aberrant usage of antibiotics, or exposure to severe enterocolitis induce gut dysbiosis, resulting in mucosal immune activation and low-grade inflammation, as well as gut barrier dysfunction through increased epithelial permeability. These pathways along with psychological factors influence the brain-gut-microbiome axis, which is mediated by aberrant neuroendocrine factors, transmitters, and metabolites, resulting in altered gut motility and visceral hypersensitivity. Chronic status of such gut dysfunctions shapes IBS symptoms including abnormal bowel movements and visceral pain. TLR, Toll-like receptor; CRH/HPA, corticotropin-releasing hormone/hypothalamic-pituitary-adrenal axis; ENS, enteric nervous system; CNS, central nervous system.

**Figure 2 ijms-21-08664-f002:**
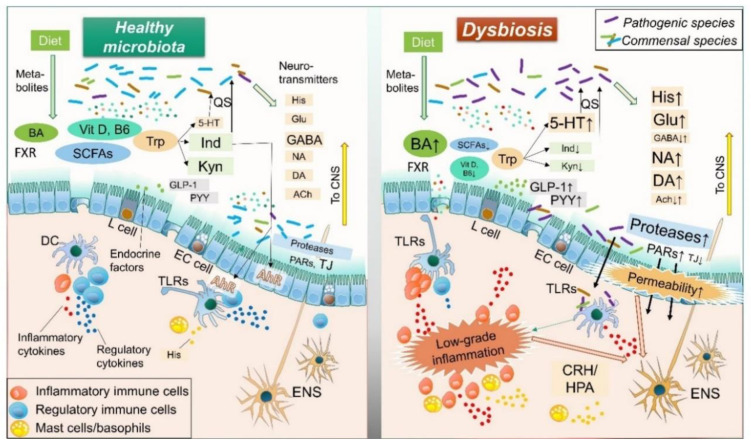
**Mechanisms of dysbiosis-mediated intestinal dysfunction in IBS.** In dysbiosis related to IBS, both the quality and quantity of microbial metabolites (BA, SCFAs, Vit D, B6, Trp-related), neurotransmitters (His, Glu, GABA, NA, DA, ACh), compounds (TLR ligands), neuroendocrine factors (GLP-1, PYY), and protases are significantly altered as compared with those in a healthy condition. In addition, activated 5-HT with weakened Ind and Kyn signaling can be observed in the Trp pathway. These alternations potentially increase mucosal permeability, while microbial products (TLR ligands, etc.) or antigens can easily penetrate into sub-mucosal areas and stimulate immune cells, thus inducing low-grade inflammation. Prolonged mucosal inflammation and some microbial products directly or indirectly influence ENS and CNS functions through CRH/HPA, GI motility alteration, and hypersensitivity as part of the pathogenesis of IBS. BA, bile acids; SCFA, short-chain fatty acids; Vit, vitamin; Trp, tryptophan; QS, quorum sensing; 5-HT, 5-hydroxytryptamine; Ind, indole; Kyn, kynurenine; His, histamine; Glu, glutamine; GABA, γ-aminobutyric acid; NA, noradrenaline; DA, dopamine; Ach, acetylcholine; GLP-1, glucagon-like peptide-1; PYY, Peptide YY; AhR, aryl hydrocarbon receptor; PARs, protease activated receptors; TJ, tight junction; TLRs, toll-like receptors; EC cells, enterochromaffin cells; CRH/HPA, corticotropin-releasing hormone/hypothalamic-pituitary-adrenal axis; ENS; enteric nervous system; CNS; central nervous system.

**Table 1 ijms-21-08664-t001:** Functions of microbial products in pathogenesis of IBS.

Microbial Products	Visceral Pain	GI Motility	Mucosal Permeability	Mucosal Inflammation	Influencing Dysbiosis	References
**Neurotransmitters**						
Histamine	↑↑	↑	↑	↑		[53,78,79,80,81,82,83,84]
Serotonin	↑	↑	↑	↑	++	[7,85,86,87,88]
Glutamate	↑↑		↓?	↑	+	[89,90,91,92,93,94,95,96,97,98,99]
γ-aminobutyric acid	↓	↓		↓		[92,100,101,102,103,104,105]
Noradrenalin	↑	↑	↑	↑↓		[106,107,108,109,110,111]
Dopamine	↓	↑↓	↑?	↓?	+	[112,113,114,115,116,117,118]
Acetylcholine		↑		↓		[111,112,119,120]
**Compounds**						
Toll-like receptor ligands	↑↓	↑↓	↑↓	↑↓		[121,122,123]
Cytokines	↑↓	↑↓	↑↓	↑↓		[52,124,125,126,127,128,129,130]
Pore-forming toxins, N-formylated peptides	↑		↑	↑		[124,131,132,133]
**Metabolites**						
Tryptophan (aryl hydrocarbon receptor, kynurenine pathways)	↓↑		↓	↓	+	[89,124,134,135,136,137,138,139,140,141]
Short-chain fatty acids	↑↓	↑	↓	↓		[70,142,143,144,145,146,147,148,149,150,151,152,153,154]
Bile acids	↓	↑		↑	+	[155,156,157,158,159,160]
Vitamin D and B6	↓		↓	↓	+	[161,162,163,164,165,166,167,168,169,170]
**Endocrine factors**						
Glucagon-like peptide-1	↓	↓		↓		[171,172,173,174,175]
Peptide YY	↓	↓				[172,176,177,178]
**Enzymes**						
Proteases	↑	↑↓	↑↑↑	↑		[124,179,180,181,182,183,184,185,186]

↑, up-regulated; ↓, down-regulated; +, positive.

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
