# Peer review of "Molecular Mechanisms of Microbiota-Mediated Pathology in Irritable Bowel Syndrome"

_ijms, 2020, doi:10.3390/ijms21228664_

Round 1

Reviewer 1 Report

A well written and exhaustive review focused on multiple aspects of Irritable bowel syndrome (IBS). In particular, the paper clearly summarizes current data coming out in favour of a pathogenetic link between this disease and gut microbiota. Authors analyzes point by point the different, complex, and today not completely understood molecular aspects microbiota-related and their possible role. When completely clarified, targeting dysbiosis as a treatment will be possible

Author Response

We thank the Reviewer 1 for the favorable comments.

Reviewer 2 Report

Authors in the article "Molecular mechanisms of microbiota-mediated pathology in irritable bowel syndrome" decided to review scientific literature on associations between gut microbiota activity and its metabolites and pathophysiology of an irritable bowel syndrome. This research area is very interesting and brings new insight into close relationship between gut microbiota and host's wellbeing.

Broad comments

Authors in presented article reviewed thoroughly role of gut microbiota and its metabolites in pathophysiology of irritable bowel syndrome. Many aspects were included concerning factors affecting prevalence of IBS, difficulties with studying this disease in humans and review of groups of metabolites involved in gut health and potential pathogenesis of IBS. This article is well written and interesting to the readers. However there are some small spelling errors and additional aspects that might be expanded to enrich value of this article.

Specific comments

  1. An additonal asset of this article is including positive and negative aspects of activity of gut microbiota. Also in lines 108-118 it is impressive that  2 aspects of correlation between dysbiosis and IBS were investigated. 

  2. Authors in lines 127-133 wrote that Lactobacillus and Bifidobacterium increase bowel motorical activity. It is an interesting aspect, because these bacteria are present in some probiotics. Might theme reveal positive impact on patients with IBS-C type? It would be an interesting aspect to invesigate. 

  3. The use of fecal microbiota transplantation was included in lines 119-121 of the review as one of promissing therapeutic options. It is an interesting novel technique with increasing number of aplications. Its positive role in IBS might be explained more thoroughly in 1-2 sentences.

  4. line 112, 139 - PI-IBS abbreviation was explained in line 139, however it was first mentioned in line 112 without explanation.

  5. Small writing errors are listed belowe for English correction:
    -line12 - gut resident microbiota appears
    -line 224-225 - "treating IBS treatment" should be rewritten
    line 186 - Tri is an abbreviation in text and on figure there is Trp 

Author Response

  1. An additonal asset of this article is including positive and negative aspects of activity of gut microbiota. Also in lines 108-118 it is impressive that 2 aspects of correlation between dysbiosis and IBS were investigated.

>We appreciate the positive comment.

  1. Authors in lines 127-133 wrote that Lactobacillus and Bifidobacterium increase bowel motorical activity. It is an interesting aspect, because these bacteria are present in some probiotics. Might theme reveal positive impact on patients with IBS-C type? It would be an interesting aspect to invesigate.

> We agree with the reviewer, but this observation was obtained by the study using germ-free mice. As we mentioned in the text, there is huge differences between germ-free and specific pathogen-free mice as well as experimental animals and humans regarding multiple aspects (new lines 49-52 and 143-145). Therefore, although interesting, we need to interpret these results carefully. Whereas these bacteria have already found to be effective for IBS patients (but not specific to IBS-C) as prebiotics (e.g. Lyra et al. Irritable bowel syndrome symptom severity improves equally with probiotic and placebo. World J Gastroenterol. 2016 Dec 28; 22(48): 10631–10642.).

  1. The use of fecal microbiota transplantation was included in lines 119-121 of the review as one of promissing therapeutic options. It is an interesting novel technique with increasing number of aplications. Its positive role in IBS might be explained more thoroughly in 1-2 sentences.

>We added the carful discussion regarding the fecal microbiota transplantation (FMT) in lines 121-128 as shown below.

“Particularly, FMT appears to be one of the most powerful modalities to reverse dysbiosis and potentially induce long-lasting remission of IBS. Recent study demonstrated that utilizing a well-defined donor with a normal dysbiosis index and favorable specific microbial signature is important for successful FMT in IBS patients. Whereas the efficacy of FMT is not always superior to placebo and several safety concerns including fatal infection have been raised. More recently, potential risk of transmission of severe acute respiratory syndrome coronavirus 2 (SARS-CoV-2) via FMT has been considered. Therefore, careful consideration of FMT should be given in case of IBS.”

  1. line 112, 139 - PI-IBS abbreviation was explained in line 139, however it was first mentioned in line 112 without explanation.

>Thank you for pointing this issue out. We revised it accordingly.

  1. Small writing errors are listed belowe for English correction:

-line12 - gut resident microbiota appears

-line 224-225 - "treating IBS treatment" should be rewritten

-line 186 - Tri is an abbreviation in text and on figure there is Trp

>We also appreciate for your check-up. We modified all of them according to the comments.

Reviewer 3 Report

The review is well written presenting the most important knowledge on the role of intestinal microbiota and different microbial compounds, metabolites, factors and enzymes in the pathogenesis of IBS.

There are only a few changes that should be done:

  • The keywords are too many, better to have 5 or 6 important keywords.
  • In Table 1 the abbreviations should be explained in a legend under the table.
  • In Figure 1 I would not use an arrow, but I would write Increase permeability; Also abbreviations should be explained (even explained already)

There is no need to improve the use of the English language, just typo check-up (eg. plasma samples instead of plasma sample - line 332)

Author Response

  1. The keywords are too many, better to have 5 or 6 important keywords.

>We have reduced the number of keywords accordingly. However, we think that those key words are helpful for readers to find this review article easily by the PubMed search, and will potentially increase the number of citations. So, we would like to show as many key words as possible if permitted.

  1. In Table 1 the abbreviations should be explained in a legend under the table.

>We modified the table including the format and added the abbreviations under the Table 1.

  1. In Figure 1 I would not use an arrow, but I would write Increase permeability; Also abbreviations should be explained (even explained already)

>We changed the words in the figure accordingly. Abbreviations are shown in the Figure legend 1.

  1. There is no need to improve the use of the English language, just typo check-up (eg. plasma samples instead of plasma sample - line 332),

>Thank you for the comment. We have checked typographical errors and modified them accordingly (new line 340).